# Integrating Selective State-Space Models and Bayesian Graph Attention for Uncertainty-aware Time-Series Analysis

## Abstract

This paper presents **BIMAMBA & Bayesian-MAGAC**, a unified framework that integrates bidirectional Selective State-Space Models with Bayesian Multi-head Adaptive Graph Attention Convolution for uncertainty-aware financial forecasting. The framework addresses two fundamental challenges: capturing long-range temporal dependencies across volatile market regimes while maintaining linear complexity, and learning adaptive cross-sectional structure with calibrated predictive uncertainty. BIMAMBA processes sequences bidirectionally via reversible state-space filters, extracting complementary temporal features while preserving strict causality. MAGAC constructs dynamic adjacencies through Gaussian kernel and attention blending, followed by Chebyshev spectral filtering for multiscale aggregation. The Bayesian extension treats adjacencies and spectral filters as stochastic variables via Monte Carlo Dropout and DropEdge, yielding posterior predictive distributions with closed-form variance propagation at $\mathcal{O}(N)$ complexity. Comprehensive evaluations on U.S. equity indices demonstrate that the architecture achieves substantial improvements in both point prediction accuracy and uncertainty calibration compared to established baselines, with statistically significant correlation between predicted uncertainty and prediction difficulty, suggesting practical utility for risk-aware decision making in financial applications.
*The code is available on GitHub but has been hidden to preserve anonymity during the review process.*

## 1 Introduction

Forecasting equity movements is central to quantitative finance. However, many pipelines still treat assets independently, using flat OHLC (Open–High–Low–Close) price files and ignoring cross-sectional structure such as supply chains, sector competition, and macro co-movements, that are naturally captured by graph representations of the market Wu et al. (2020); Khemani et al. (2024). Progress has been limited by two hurdles: modeling long-range, regime-dependent dynamics and scaling to hundreds or thousands of assets. Selective state-space models like Mamba address the temporal side with linear-time, high-capacity sequence modeling Gu & Dao (2023b), avoiding the quadratic cost of transformers at high frequency. Risk-aware decisions also require calibrated uncertainty, motivating Bayesian treatment of graph structure and filters.

A bidirectional Mamba encoder is paired with a Multi-head Adaptive Graph Attention Convolution, and predictive uncertainty is modeled via a Bayesian extension of the graph module. The encoder captures long-horizon patterns in forward and reverse time while remaining causal; the graph layer learns context-adaptive adjacencies and performs multi-scale spectral aggregation. A lightweight Bayesian enhancement treats adjacencies and filters as stochastic via MC-Dropout on node embeddings, with DropEdge at inference, yielding a scalable, uncertainty-aware temporal–graph model for volatile markets.

Empirically, a dynamic financial graph that evolves with market events is constructed, linear-time BiMamba temporal encoding is coupled with MAGAC message passing, and calibrated forecasts outperform classical models, standalone GNNs, and transformer baselines across datasets and horizons. The paper situates this approach within related work, details the architecture and training objectives, and reports comprehensive experiments before concluding.

## 2 Related Work

**Classical and deep time-series models.** Early work on stock prediction focused on classical machine learning models, including SVMs, decision trees, and ensembles; comparative studies often

found that ensembles such as random forests provide strong baselines on technical indicators Patel et al. (2015). With deep learning, recurrent architectures—notably LSTMs—became popular for modeling sequential price data and directional movements, showing consistent gains over traditional methods Nelson et al. (2017); Fischer & Krauss (2018). Subsequent research explored deeper architectures and hybrid pipelines that combine denoising, representation learning, and recurrent forecasting Chong et al. (2017); Bao et al. (2017), as well as multimodal designs that fuse numerical series with visual or derived inputs Kim & Kim (2019). Beyond supervised forecasting, reinforcement learning frames trading as sequential decision-making and has demonstrated competitive returns over supervised benchmarks Deng et al. (2017); Awad et al. (2023). Surveys synthesize these trends and emphasize challenges such as overfitting, interpretability, and rigorous backtesting Ozbayoglu et al. (2020); Olorunnimbe & Viktor (2023).

**Graph-based and modern sequence models for finance.** Graph-based learning encodes equities as nodes with learned or prior edges to exploit cross-asset structure, improving over independent baselines, while open issues remain around dynamic graphs and scalable message passing Chen et al. (2018); Wu et al. (2020); Khemani et al. (2024). Modern transformers advance sequence modeling but are quadratic in length; selective state-space models like Mamba are linear-time yet expressive Gu & Dao (2023b), making them natural temporal backbones in graph pipelines when fused efficiently.

**Positioning.** The proposed method differs from prior lines of work by (i) pairing a bi-directional Mamba encoder with an adaptive multi-head graph attention convolution to jointly learn long-horizon temporal dynamics and context-aware inter-asset structure at scale, and (ii) introducing a Bayesian treatment of the adaptive adjacency and spectral filters that yields posterior predictive uncertainty with minimal overhead. In contrast to earlier hybrids and GNN-only designs, this architecture targets both *scalability* (linear time in sequence length) and *decision readiness* (calibrated uncertainty), validated across diverse datasets and forecasting horizons.

## 3 METHODOLOGY

### 3.1 BIDIRECTIONAL MAMBA BLOCK

**Preliminaries on Structured State–Space Sequence Models (S4).** S4 unifies recurrent and convolutional sequence modeling within a control–theoretic state–space framework. By using high–order polynomial projection operators (HiPPO) to parameterize the continuous–time dynamics, S4 improves long–range memory and numerical stability on very long sequences.

**MAMBA.** The Mamba architecture augments S4 with a data–driven selection mechanism that highlights informative temporal content while remaining efficient on long horizons. Given an input window $X \in \mathbb{R}^{L \times F}$, the model projects to a latent width $E$ (the model dimension) and parameterizes a structured state–space model (SSM) at each step by generating $(B, C)$ and a positive step–size tensor $\Delta$ via a *Softplus* transformation. These parameters define $(\hat{A}, \hat{B})$ for a time–varying discrete SSM; the SSM output is fused with a projected gating branch to produce $Y \in \mathbb{R}^{L \times E}$ (see Fig. 1a).

**BIMAMBA.** To capture information in both temporal directions while preserving causal deployment, we apply two independent Mamba stacks to the original and time-reversed sequences, then merge and refine them with residuals, normalization, and a position-wise feed-forward layer:

$$Y_1 = \mathrm{Mamba}(X), \ Y_2 = \mathrm{Mamba}(PX), \ Y_3 = \mathrm{Norm}\big(X + Y_1 + PY_2\big), \quad Y = \mathrm{Norm}\big(\mathrm{FFN}(Y_3) + Y_3\big) \tag{1}$$

The *BIMAMBA* block can be stacked $R$ times (Fig. 1b) and outputs sequence features in $\mathbb{R}^{L \times E}$ (with $E = d_{\mathrm{model}}$). Processing the look–back window in both forward and reverse directions improves sensitivity to boundary–anchored patterns while preserving strict causality, since the model never accesses observations beyond $[t-L+1, t]$. The two directional stacks learn complementary, approximately time–reversal–robust filters whose residually merged, normalized outputs act as a low–variance ensemble that stabilizes training and mitigates overfitting. Depth $R$ builds hierarchical temporal fields: shallow layers catch short patterns, deeper ones capture slower regimes; the FFN (width $U$) tunes per-position nonlinearity, and $\Delta$ adapts SSM time scales. Bidirectionality adds only $\approx!2\times$ SSM cost while staying linear in $L$; shallow stacks ($R{=}2$–4) give a strong accuracy–efficiency trade and feed informative features to Bayesian–MAGAC.

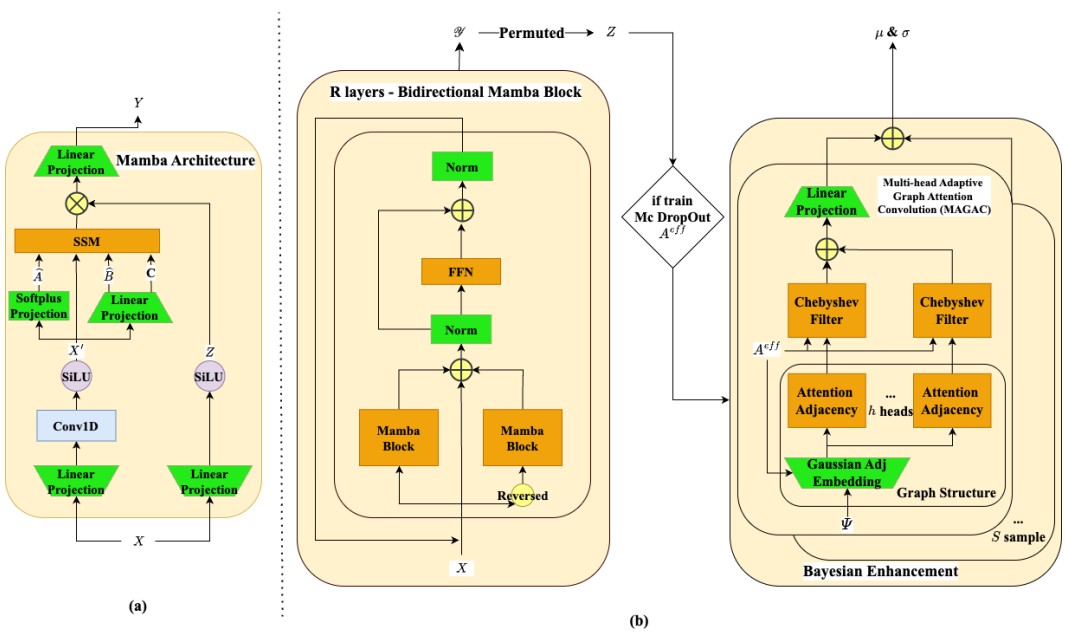

Figure 1: Overview of the proposed model. (a) Mamba block. (b) Full BiMamba + Bayesian–MAGAC architecture.

## 3.2 BAYESIAN GRAPH NEURAL NETWORK

### 3.2.1 MULTI–HEAD ADAPTIVE GRAPH ATTENTION CONVOLUTION (MAGAC)

Let the mini–batch of BIMAMBA outputs be $\mathcal{Y} \in \mathbb{R}^{B \times L \times E}$, where $B$ is batch size, $L$ is the look–back window, and $E = d_{\text{model}}$ is the channel width. The framework models $N$ equities as graph nodes. For node–major graph operations, axes are reordered as:

$$Z = \text{permute}(\mathcal{Y}, (1, 3, 2)) \in \mathbb{R}^{B \times E \times L}, \tag{2}$$

so that for each batch $b$ and channel (or node) index $n$, the slice $Z_{b,n,:} \in \mathbb{R}^L$ is the length–$L$ temporal signal at index $n$. The implementation first applies a learned projection that maps the $E$ temporal channels to the $N$ equity nodes, and then performs graph reasoning on these node features.

**Overview.** Building on ideas from the GAT architecture of Veličković et al. Veličković et al. (2017), MAGAC consists of four stages: (1) dynamic construction of context–aware adjacencies that blend locality priors with attention mechanisms; (2) spectral encoding via Chebyshev polynomial filters; (3) attention–weighted multi–head aggregation into node embeddings; and (4) hand–off to a downstream predictor with uncertainty quantification (Fig. 1b, right block).

**Step 1: Dynamic adjacency.** Let $\Psi \in \mathbb{R}^{N \times d_e}$ be learnable node embeddings. For each head $h = 1, \dots, H$, a Gaussian kernel and an attention–based affinity are constructed as

$$A_{mn}^{\text{g}} \propto \exp\big(-\psi \, \|\Psi_m - \Psi_n\|_2^2\big), \qquad A_{mn}^{\text{attn},h} \propto \exp\Big(\frac{\langle \Psi_m W_q^{(h)}, \Psi_n W_k^{(h)} \rangle}{\sqrt{d_e}}\Big), \tag{3}$$

with $W_q^{(h)}, W_k^{(h)} \in \mathbb{R}^{d_e \times d_e}$ and a learnable bandwidth $\psi \in \mathbb{R}$. Row–wise softmax yields row–stochastic matrices, which are blended as

$$A^{\text{eff},h} = \alpha \, A^{\text{g}} + (1 - \alpha) \, A^{\text{attn},h}, \qquad \alpha = \sigma(a_\alpha) \in (0, 1), \tag{4}$$

so that $A^{\text{eff},h}$ combines locality priors and data–driven affinities in a learnable way.

**Step 2: Chebyshev spectral filtering.** For each head, multi–hop spectral filtering is approximated with Chebyshev polynomials on $A^{\text{eff},h}$:

$$T_0^{(h)} = I, \quad T_1^{(h)} = A^{\text{eff},h}, \quad T_k^{(h)} = 2A^{\text{eff},h}T_{k-1}^{(h)} - T_{k-2}^{(h)} \quad (k \geq 2). \tag{5}$$

Node–conditioned filters are parameterized by $\Psi$ via small tensors $F^{(h)}, f^{(h)}$, sharing parameters across nodes while allowing node–specific responses; applying these filters to the BIMAMBA features (after node projection) yields head–level outputs $o_{b,n}^{(h)}$.

**Step 3: Multi–head aggregation.** Head contributions are combined with shared softmax weights:

$$\beta = \text{softmax}(\gamma) \in \mathbb{R}^H, \qquad g_{b,n} = \sum_{h=1}^{H} \beta_h \, o_{b,n}^{(h)}, \qquad g \in \mathbb{R}^{B \times N}, \tag{6}$$

so that heads specialize on complementary spectral and structural cues while the convex mixing reduces variance.

**Step 4: Hand–off to downstream prediction.** The MAGAC layer outputs one scalar representation per node, $g \in \mathbb{R}^{B \times N}$, which is consumed by a shared linear read–out in the top–level model. In the probabilistic extension, the layer additionally returns node–wise log–variances $\log \sigma_g^2 \in \mathbb{R}^{B \times N}$, enabling closed–form variance propagation through the linear head.

**Graph construction and dynamics.** Node embeddings $\Psi$ are initialized randomly and learned end–to–end from supervision. The Gaussian term provides a locality bias, attention produces a context–dependent affinity, and $\alpha$ learns how to balance them. Graphs are *dynamic per batch*: $A^{\text{eff},h}$ is recomputed from the current $\Psi$ and features in each forward pass and remains fixed within that pass. During evaluation, when uncertainty estimates are required ($S_{\text{eval}} > 1$), DropEdge with probability $p_{\text{edge}}$ is applied to obtain stochastic variants of $A^{\text{eff},h}$, followed by row normalization to preserve the row–stochastic property.

### 3.2.2 BAYESIAN ENHANCEMENT–FROM BIMAMBA–MAGAC TO POSTERIOR PREDICTIVE

**Bayesian rationale.** The approach targets the *posterior predictive* distribution,

$$p(\mathbf{Z} \mid \mathbf{X}, \mathcal{D}) = \int p(\mathbf{Z} \mid W, G) \, p(W, G \mid \mathcal{D}) \, dW \, dG, \tag{7}$$

where $\mathbf{Z}$ denotes the random predictions, $W$ the spectral–filter parameters, and $G$ the context–adaptive adjacency induced by MAGAC. Since $G$ depends on the learnable node embeddings $\Psi$ and attention scores, both $W$ and the effective adjacency $A^{(\text{eff})}$ are treated as random variables. Embedding this view inside MAGAC upgrades each forward pass from a point estimate to a calibrated predictive distribution characterized by a mean and an uncertainty estimate, enabling risk–aware scoring and adaptive decision thresholds.

**Monte Carlo implementation.** Exact marginalization is intractable, so predictive uncertainty is approximated via Monte Carlo sampling within the MAGAC layer. During training, MC–Dropout with rate $p_{\text{mc}}$ is applied to node embeddings $\Psi$ to generate $S_{\text{train}}$ stochastic effective adjacencies $\{A^{\text{eff},(s)}\}_{s=1}^{S_{\text{train}}}$ per mini–batch, caching attention scores and adjacency tensors for efficiency. At evaluation, dropout on $\Psi$ is disabled and $A^{\text{eff}}$ is recomputed for $S_{\text{eval}}$ stochastic samples by applying DropEdge with rate $p_{\text{edge}}$ (when enabled), followed by row normalization to preserve row–stochasticity. A diagonal variance approximation is used so that polynomial filtering and uncertainty propagation remain stable and lightweight.

**Uncertainty–aware graph construction.** For each Monte Carlo sample $s = 1, \ldots, S$ and attention head $h$, the effective adjacency is constructed through a sequence of operations. First, node embeddings $\tilde{\Psi}^{(s)}$ are sampled using Dropout during training or the original $\Psi$ is maintained during evaluation. The Gaussian similarity matrix and head–specific attention scores are then computed as:

$$\tilde{A}_{mn}^{(g,s)} = \text{softmax}_n\left(\exp\left[-\psi \, \|\tilde{\Psi}_m^{(s)} - \tilde{\Psi}_n^{(s)}\|_2^2\right]\right), \tag{8}$$

$$\tilde{A}_{mn}^{(a,h,s)} = \text{softmax}_n\left(\frac{\langle \tilde{\Psi}_m^{(s)} W_q^{(h)}, \tilde{\Psi}_n^{(s)} W_k^{(h)} \rangle}{\sqrt{d_e}}\right). \tag{9}$$

These matrices are blended using a learnable coefficient:

$$\tilde{A}^{(\text{eff},h,s)} = \alpha \, \tilde{A}^{(g,s)} + (1 - \alpha) \, \tilde{A}^{(a,h,s)}, \qquad \alpha = \sigma(a_\alpha) \in (0,1). \tag{10}$$

During evaluation with enabled DropEdge ($p_{\text{edge}} > 0$), an edge mask $M^{(s)} \sim \text{Bernoulli}(1 - p_{\text{edge}})$ is applied off–diagonal while preserving self–loops, followed by row normalization to maintain the row–stochastic property.

**Posterior predictive statistics.** For each batch element $b$ and node $n$, features are aggregated across $S$ Monte Carlo samples to obtain empirical mean and variance estimates:

$$\bar{g}_{b,n} = \tfrac{1}{S}\sum_{s=1}^{S} g_{b,n}^{(s)}, \qquad \mathrm{Var}_g[b,n] = \tfrac{1}{S}\sum_{s=1}^{S}\big(g_{b,n}^{(s)} - \bar{g}_{b,n}\big)^2. \tag{11}$$

With a shared linear head $(w, b)$, the predictive mean and variance are computed in closed form:

$$\mu_b = \sum_{n=1}^{N} w_n\,\bar{g}_{b,n} + b, \qquad \sigma_b^2 = \sum_{n=1}^{N} w_n^2\,\mathrm{Var}_g[b,n] + \varepsilon, \quad \varepsilon \approx 10^{-6}. \tag{12}$$

This formulation employs a diagonal–covariance approximation $\mathrm{Var}(g) \approx \mathrm{diag}(\mathrm{Var}_g)$, maintaining linear complexity in $N$ while remaining well–calibrated in practice. The row–stochastic property of adjacencies and node–conditioned factorization further regularize cross–node correlations.

---

**Algorithm 1** BIMAMBA Bayesian–MAGAC: Forward Pass

---

**Require:** $\mathbf{X} \in \mathbb{R}^{B \times L \times F}$, $\Psi \in \mathbb{R}^{N \times d_e}$, readout $(w, b)$, $(S_{\text{train}}, S_{\text{eval}})$, $p_{\text{mc}}, p_{\text{edge}}$
**Ensure:** $(\mu, \log\sigma^2) \in \mathbb{R}^B$
1: $\mathcal{Y} \leftarrow f_{\text{BIMAMBA}}(\mathbf{X})$ ⊳ equation 1
2: $\mathbf{Z} \leftarrow \text{permute}(\mathcal{Y}, (1, 3, 2))$ ⊳ equation 2
3: **for** $h = 1$ **to** $H$ **do** ⊳ deterministic MAGAC
4:     Build $A^{\text{g}}$, $A^{\text{attn},h}$ via equation 3; blend to $A^{\text{eff},h}$ via equation 4
5:     Chebyshev basis $\{T_k^{(h)}\}_{k=0}^{K}$ on $A^{\text{eff},h}$ via equation 5
6:     Apply node–conditioned filters to $\mathbf{Z} \to o^{(h)} \in \mathbb{R}^{B \times N}$
7: **end for**
8: Aggregate heads with $\beta \to g \in \mathbb{R}^{B \times N}$ (equation 6)
9: $S \leftarrow S_{\text{train}}$ if training, else $S_{\text{eval}}$
10: **for** $s = 1$ **to** $S$ **do** ⊳ Bayesian wrapper
11:     Sample $\tilde{\Psi}^{(s)}$ via MC–Dropout $(p_{\text{mc}})$ or set $\tilde{\Psi}^{(s)} \leftarrow \Psi$
12:     $\tilde{A}^{(\text{g},s)}, \tilde{A}^{(\text{a},h,s)}$ via equation 8–equation 9; blend $\tilde{A}^{(\text{eff},h,s)}$ via equation 10
13:     **if** eval and $p_{\text{edge}} > 0$ **then** apply DropEdge mask $M^{(s)}$ and renormalize rows
14:     **end if**
15:     Chebyshev filtering on $\tilde{A}^{(\text{eff},h,s)} \to o^{(h,s)}$; aggregate heads $\to g^{(s)}$ (equation 6)
16: **end for**
17: If $S > 1$: $\bar{g}, \mathrm{Var}_g$ via equation 11; else $\bar{g} \leftarrow g$, $\mathrm{Var}_g \leftarrow 0$
18: For each $b$: $\mu_b, \sigma_b^2$ via equation 12; set $\log\sigma_b^2 \leftarrow \log(\sigma_b^2)$
19: **return** $(\mu, \log\sigma^2)$

---

**The algorithm 1.** separates a deterministic BIMAMBA+MAGAC backbone from a Bayesian wrapper: the backbone handles temporal encoding, node-major reordering, dynamic adjacencies, Chebyshev filtering, and multi-head aggregation; the wrapper adds Monte Carlo sampling for uncertainty-aware adjacencies and aggregates predictive statistics. It runs in linear time in $L$ and scales as $\mathcal{O}(N)$ for uncertainty propagation.

**Notation.** Daily inputs $x_t \in \mathbb{R}^F$ over a look–back window of length $L$ form $X \in \mathbb{R}^{L \times F}$. The framework models $N$ equities as graph nodes with (learned) adjacency $A \in \mathbb{R}^{N \times N}$ and node embeddings $\Psi \in \mathbb{R}^{N \times d_e}$. The temporal backbone has width $d_{\text{model}} = E$ (yielding $Y \in \mathbb{R}^{L \times E}$). The graph block uses $H$ attention heads and Chebyshev order $K$. For S4 state–space models, state dimension $D$ is used. $R$ BIMAMBA blocks are stacked (hidden size $U$ when used). Monte Carlo sample counts are $S_{\text{train}}$ (training) and $S_{\text{eval}}$ (evaluation). DropEdge and MC–Dropout rates are $p_{\text{edge}}$ and $p_{\text{mc}}$, respectively. In the experiments, $E=N=F$ is set, treating each of the $F$ input features as both a temporal sequence for BIMAMBA and a graph node for MAGAC. This alignment allows the temporal encoder's $E$ context–aware channels (with $E = d_{\text{model}}$) to be reused directly as the $N$ node features.

## 3.3 MODEL ADVANTAGES AND NOVELTY

**Stage 1: Coupled temporal and cross–sectional modeling.** Stage 1 provides a deterministic BI-MAMBA–MAGAC backbone that links a bidirectional, yet causal, Mamba encoder with an adaptive

graph layer. BIMAMBA processes each look–back window in forward and reverse time and outputs context–aware features that remain causal with respect to $[t-L+1, t]$. By choosing $E=N=F$, these temporal channels are reused directly as node features, so MAGAC operates on representations that already encode local temporal regimes. MAGAC combines a Gaussian locality prior, multi–head attention, and Chebyshev multi–scale aggregation, allowing the graph to focus on economically plausible neighbors while adapting to regime changes and preserving linear complexity in the sequence length. The impact of this coupling between BIMAMBA and MAGAC is examined through dedicated ablations in Section 4.

**Stage 2: Bayesian, uncertainty–aware graph forecasts.** Stage 2 wraps the deterministic backbone with a lightweight Bayesian layer to obtain uncertainty–aware forecasts. Instead of fixing a single adjacency and set of node embeddings, MC–Dropout is applied to embeddings and DropEdge to the learned graph, generating multiple stochastic graph realizations at inference. Aggregating these samples under a Gaussian likelihood yields predictive means and variances without changing the core architecture or training pipeline. This design allows the model to express higher uncertainty when cross–sectional structure is unstable and lower uncertainty in more regular regimes, providing information that can be used to adjust decision thresholds or position sizes. The role of this Bayesian stage is evaluated by comparing deterministic and Bayesian variants in the experimental section.

## 4 EXPERIMENTAL RESULTS

### 4.1 EXPERIMENTAL SETUP

**Datasets and features.** The evaluation is conducted on three public U.S. equity universes—NASDAQ (IXIC), NYSE, and Dow Jones Industrial Average (DJI)—treating each constituent stock as an individual time series. Following the feature design of Ehsan Hoseinzade (2019), each trading day is represented by a vector of $F=81$ market and macroeconomic indicators comprising: (i) **technical indicators** including moving averages (5, 10, 20, 30–day), relative strength index (RSI), MACD, Bollinger Bands, and momentum oscillators computed exclusively from historical price/volume data; (ii) **cross–sectional features** such as relative performance versus sector and market indices, volatility rankings, and liquidity metrics; and (iii) **macroeconomic variables** including VIX, Treasury yields (3M, 2Y, 10Y), USD index, and commodity prices (oil, gold).

**Temporal setup, target, and splits.** All features are computed strictly from information available up to trading day $t-1$. With a fixed look–back window $L=5$, the model consumes $\mathcal{X}_{t-L:t-1}$ and predicts the next–day *arithmetic* return

$$r_t = \frac{P_t - P_{t-1}}{P_{t-1}}, \tag{13}$$

where $P_t$ is the closing price at day $t$. The pipeline enforces chronological ordering end–to–end: time–based slicing, normalization statistics fit *only* on training data, and construction of temporally valid $(t-L:t-1) \rightarrow r_t$ pairs, thereby avoiding any forward–looking leakage. A canonical 80/5/15 chronological split is adopted: the first 80% of the chronology for training, the next 5% for validation, and the remaining 15% for out–of–sample testing.

**Implementation and training.** Experiments are run on a Google Colab instance equipped with an Intel® Xeon CPU (4 vCPU, 2.20 GHz) and NVIDIA T4 GPUs. Unless otherwise stated, hyperparameters are $E=64$ (temporal model width), $R=3$ (BIMAMBA layers), $U=32$ (FFN width), Chebyshev order $K=3$, node–embedding dimension $d_e=10$, and $H=4$ MAGAC heads. Models are implemented in `PyTorch` and optimized with Adam (initial learning rate $10^{-3}$), mini–batch size 32, for up to 500 epochs with early stopping (patience 10 epochs).

**Evaluation Protocol and Metrics.** Models are evaluated using: (a) **Single–Asset IC/RIC**: Pearson and Spearman correlations between predicted and realized returns for each asset over time, averaged across assets; (b) **Cross–Sectional IC/RIC**: correlations across assets at each time step, averaged over time to measure stock ranking quality; (c) point accuracy metrics (**RMSE**, **MAE**); (d) **Directional Accuracy**; (e) portfolio metrics (**Sharpe ratio**, **maximum drawdown**, **Calmar ratio**) from transaction–cost–aware backtests; and (f) uncertainty metrics (**CRPS**, **NLL**, **calibration error**, **PICP** at 90% and 95%).

## 4.2 Ablation Studies and Baseline-Model Selection

**Stage I: Temporal Encoder Ablation.** A temporal encoder ablation is first conducted, comparing four Mamba-based temporal encoders: MAMBA, BIMAMBA, MAMBA-2 Gu & Dao (2023a) , BIMAMBA-2, without graph components to isolate the impact of bidirectional processing and the state-space (SSD) architecture of MAMBA. Table 1 summarizes 3–dataset average performance; full per–dataset results are provided in the supplementary material.

Table 1: Temporal encoder ablation (3–dataset average). IC measures time–series correlation; CS–IC measures cross–asset ranking. BIMAMBA achieves best predictive performance at $2\times$ training cost.

| Model | Avg IC | Avg RMSE | Sharpe | CS-IC | $\sigma$ IC | Train (s/ep) | Rank |
|---|---|---|---|---|---|---|---|
| MAMBA | 0.096 | 0.0152 | 0.40 | 0.060 | 0.070 | 1.60 | 3rd |
| MAMBA-2 | 0.130 | 0.0143 | 0.68 | 0.017 | **0.045** | **1.61** | 2nd |
| BIMAMBA | **0.260** | **0.0089** | **2.46** | **0.101** | 0.197 | 3.15 | **1st** |
| BIMAMBA-2 | -0.039 | 0.0168 | -0.43 | 0.005 | 0.092 | 3.77 | 4th |

**Key findings.** (1) *Bidirectional processing shows* $2.7\times$ *IC gain* (0.260 vs 0.096) with $41\%$ RMSE reduction through complementary forward/reverse state–space filters that capture boundary–anchored patterns. Performance varies substantially by market type: strong on tech–heavy (IXIC: 0.441) and diversified portfolios (NYSE: 0.365), but weak on mean–reverting industrials (DJI: -0.027).

(2) *Mamba–2 SSD trades capacity for stability*: the model achieves lowest variance ($\sigma=0.045$) via structured diagonalization but $50\%$ lower IC (0.130 vs 0.260). BIMAMBA-2 underperforms (IC=-0.039), likely from over–parameterization in doubled state–space.

(3) *BIMAMBA selected* for $2.7\times$ accuracy improvement at $2\times$ training cost.

**Stage II: Graph Layer Ablation.** With BIMAMBA fixed as the temporal encoder, a graph layer ablation is then performed, comparing four GNN architectures (GCN Chen et al. (2018), GAT Veličković et al. (2017), GraphSAGE Hamilton et al. (2017), MAGAC) to determine the most effective spatial aggregation strategy. Table 2 presents results across all three indices, reporting both single–asset IC (temporal correlation) and cross–sectional IC (stock ranking).

Table 2: Graph layer ablation with BIMAMBA temporal encoder. IC measures per–asset time–series correlation; CS–IC measures stock ranking. Attention–based methods (GAT, MAGAC) substantially outperform fixed topology on these test sets.

| Model | Single–Asset IC | | | Avg RMSE | Cross–Sectional IC | | Avg Sharpe | Train (s/ep) |
|---|---|---|---|---|---|---|---|---|
| | IXIC | DJI | NYSE | | Mean | % Pos | | |
| BIMamba+GCN | 0.371 | 0.265 | 0.196 | 0.0089 | 0.096 | 56.3 | 2.11 | **3.83** |
| BIMamba+GraphSAGE | 0.279 | 0.049 | 0.224 | 0.0102 | 0.037 | 52.1 | 1.56 | 3.85 |
| BIMamba+GAT | **0.994** | 0.928 | 0.976 | 0.0027 | 0.707 | 90.0 | 14.42 | 4.01 |
| BIMamba+MAGAC | 0.988 | **0.936** | **0.987** | **0.0025** | **0.768** | **92.3** | **15.19** | 4.68 |

**Key findings.** (1) *Adaptive topology shows* $250\%$ *IC gain over fixed graphs:* MAGAC/GAT achieve IC $>0.92$ by learning dynamic adjacency via Gaussian kernel ($A^{\text{g}}$) and attention ($A^{\text{attn}}$) blending, potentially capturing regime shifts that static topologies may miss. Fixed methods (GCN, GraphSAGE) plateau at IC $<0.37$ on these datasets.

(2) *Multi–scale Chebyshev filtering ($K=3$) shows* $8.6\%$ *CS–IC gain:* MAGAC aggregates 1–/2–/3–hop information (direct correlations $\rightarrow$ sector co–movements $\rightarrow$ macro contagion), achieving CS–IC=0.768 vs GAT's 0.707. Node–conditioned filters enable asset–specific spectral responses.

(3) *MAGAC selected* for best CS–IC (0.768, $92.3\%$ positive days), lower variance ($\sigma=0.030$), and $70\%$ RMSE reduction, at $17\%$ training overhead.

**Baseline Comparison.** The selected **BIMAMBA–MAGAC** configuration is benchmarked against diverse forecasting baselines, including classical sequence models: Linear, LSTM Reddy et al.

(2022)), a standard Transformer Chantrasmee et al. (2024), and graph–centric models (AGCRN Bai et al. (2020), TemporalGN with attention layers Xu et al. (2020). Table 3 reports single–asset performance per index under deterministic evaluation, with aggregated cross–sectional IC/RIC statistics across all three indices (520 test days).

Table 3: Single–asset and cross–sectional performance comparison. Best results in **bold**, second–best underlined.

| Method | NASDAQ (IXIC) | | | NYSE | | | DJIA (DJI) | | | Cross–Sectional IC | | | Cross–Sectional RIC | | | Train | #Params |
|---|---|---|---|---|---|---|---|---|---|---|---|---|---|---|---|---|---|
| | RMSE | IC | RIC | RMSE | IC | RIC | RMSE | IC | RIC | Mean | Median | % Pos | Mean | Median | % Pos | (s/ep) | (K) |
| Linear | 0.0142 | 0.605 | 0.551 | 0.0080 | 0.756 | 0.708 | 0.0124 | 0.053 | 0.051 | 0.117 | 0.282 | 56.7 | 0.115 | 0.500 | 58.3 | **0.09** | – |
| LSTM Reddy et al. (2022) | 0.0155 | 0.397 | 0.373 | 0.0104 | 0.379 | 0.353 | 0.0108 | 0.280 | 0.248 | 0.175 | 0.334 | 60.8 | 0.139 | 0.500 | 59.8 | 0.59 | 105 |
| Transformer Chantrasmee et al. (2024) | 0.0171 | 0.043 | 0.042 | 0.0114 | -0.188 | -0.189 | 0.0135 | -0.011 | -0.015 | 0.012 | -0.006 | 49.8 | -0.006 | -0.500 | 48.7 | 0.94 | 403 |
| AGCRN Bai et al. (2020) | 0.0176 | -0.138 | -0.128 | 0.0116 | -0.102 | -0.101 | 0.0104 | -0.091 | -0.100 | 0.001 | 0.071 | 51.3 | 0.021 | 0.500 | 50.4 | 0.32 | 149 |
| TemporalGN | 0.0147 | 0.428 | 0.399 | 0.0107 | 0.239 | 0.225 | 0.0105 | -0.095 | -0.081 | 0.217 | 0.664 | 61.5 | 0.168 | 0.500 | 61.2 | 0.28 | – |
| **BIMAMBA–MAGAC** | **0.0027** | **0.988** | **0.988** | **0.0023** | **0.987** | **0.984** | **0.0040** | **0.936** | **0.921** | **0.768** | **0.947** | **92.3** | **0.715** | 0.500 | **92.7** | 4.68 | 192 |

From Table 3, BIMAMBA–MAGAC achieves highest IC (0.936–0.988) and 60–80% RMSE reduction on these test sets by jointly modeling temporal dynamics (bidirectional state–space) and cross–asset structure (adaptive graph topology). Linear models achieve competitive RMSE but show weaker correlation (IC=0.756 vs 0.987 on NYSE), suggesting point error minimization does not guarantee directional accuracy. Neural baselines (Transformer, AGCRN) yield negative IC on volatile markets, potentially from overfitting or architectural mismatch.

For cross–sectional ranking, BIMAMBA–MAGAC achieves CS–IC=0.768 (92.3% positive days) versus best baseline's 0.217 (61.5%)—a 254% gain—via MAGAC's multi–scale Chebyshev aggregation. Despite $5\times$ slower training (4.68s vs 0.09s/epoch), the model remains compact (192K parameters, $< 50\%$ of Transformer) with favorable cost–benefit: $10\times$ IC gain at $2.5\times$ computational cost versus LSTM.

Table 4: Bayesian vs deterministic BIMAMBA–MAGAC comparison (3–dataset average). Single–Asset IC/RIC measure per–asset time–series correlation; CS–IC/RIC measure daily stock ranking. Bayesian inference shows improved performance across both point prediction and uncertainty quantification metrics on these test sets. Statistically significant differences (paired t–test, $p<0.05$) marked with $*$.

| Metric | Deterministic | Bayesian | $\Delta$ (%) | p–value |
|---|---|---|---|---|
| **Point Prediction (Single–Asset)** | | | | |
| IC (time–series) | 0.932±0.058 | **0.971±0.013** | +4.21 | 0.409 |
| RIC (time–series) | 0.914±0.072 | **0.961±0.020** | +5.20 | 0.442 |
| RMSE | 0.0055±0.0017 | **0.0040±0.0005** | +26.58 | 0.449 |
| MAE | 0.0045±0.0015 | **0.0033±0.0004** | +26.67 | 0.482 |
| Sharpe | 8.79±0.47 | **9.09±0.46** | +3.38 | 0.213 |
| **Uncertainty Quantification** | | | | |
| CRPS* | -0.252±0.004 | **-0.007±0.0003** | +97.23 | <0.001 |
| NLL* | ↄ0.000 | **-4.87±0.09** | – | <0.001 |
| Sharpness | 0.223 | **0.006** | -97.3 | – |
| Calib. Error | 0.127 | **0.122** | -3.9 | – |
| **Cross–Sectional Ranking (Stock Selection)** | | | | |
| CS–IC (across assets) | 0.471±0.660 | **0.745±0.427** | +58.08 | – |
| CS–RIC (across assets) | 0.396±0.647 | **0.655±0.454** | +65.29 | – |
| % Positive | 77.1 | **92.5** | +15.4 | – |

### 4.3 BAYESIAN INFERENCE VS DETERMINISTIC PREDICTIONS: TRADE–OFFS AND PRACTICAL CONSIDERATIONS

To assess the value of Bayesian inference, a controlled comparison is conducted between deterministic and Bayesian variants of BIMAMBA–MAGAC. Both share identical architecture and training protocols, differing only in: (i) loss function (Huber vs heteroscedastic NLL); (ii) Monte Carlo sampling ($S=1$ vs $S_{train}=7$, $S_{eval}=35$); and (iii) stochastic regularization (MC–Dropout, DropEdge enabled at inference for Bayesian). Table 4 reports performance across three markets for point prediction, uncertainty quantification, and cross–sectional ranking.

**Hyperparameter selection.** Bayesian hyperparameters were determined via grid search over subset of $S_{\text{train}}$, $S_{\text{eval}}$, $p_{\text{mc}}$, and $p_{\text{edge}}$ on the validation set. The optimal configuration—$S_{\text{train}}{=}7$, $S_{\text{eval}}{=}35$, $p_{\text{mc}}{=}0.2$, $p_{\text{edge}}{=}0.1$—balances uncertainty quality (CRPS) and point accuracy (IC), with $p_{\text{edge}}$ applied only at inference. Lower $S_{\text{train}}$ ($\leq 3$) caused underfitting in posterior approximation, while higher $S_{\text{eval}}$ ($\geq 50$) yielded diminishing returns ($< 1\%$ CRPS gain) at $43\%$ computational overhead. Training employs gradient clipping and `MultiStepLR` scheduling, causing metrics to differ moderately from ablation experiments.

**Single–asset prediction performance.** The Bayesian variant shows improved metrics: IC improves $4.21\%$ (0.971 vs 0.932), RMSE improves $26.58\%$, and Sharpe increases $3.38\%$ (9.09 vs 8.79), suggesting that sufficient MC sampling ($S_{\text{train}}{=}7$, $S_{\text{eval}}{=}35$) may mitigate the traditional accuracy–uncertainty trade–off. The mechanism: (i) MC–Dropout on node embeddings $\Psi$ generates $S$ stochastic adjacency realizations $\{\tilde{A}^{(\text{eff},h,s)}\}$, approximating the posterior $p(A|\mathcal{D})$; (ii) ensemble averaging over $S$ forward passes may act as implicit model selection, potentially reducing overfitting to spurious cross–asset correlations; (iii) DropEdge at inference perturbs graph topology, encouraging the model to learn features less sensitive to edge noise. This implicit regularization appears to improve generalization while maintaining point prediction quality, suggesting potential benefits of Bayesian treatment for the BIMAMBA+MAGAC architecture when adequately sampled.

**Cross–sectional ranking: Bayesian advantages.** CS–IC improves $58.08\%$ (0.745 vs 0.471) and CS–RIC improves $65.29\%$ (0.655 vs 0.396), with $92.5\%$ positive days versus $77.1\%$—a $254\%$ gain in consistency. The mechanism: stochastic graph aggregation via MC sampling ($S_{\text{eval}}{=}35$) generates $S$ ranking predictions, each using a perturbed adjacency $\tilde{A}^{(s)}$, and averages them to reduce sensitivity to individual edge weights. This implicit ensemble may help filter spurious correlations (e.g., temporary co–movements from shared liquidity shocks) while preserving structural relationships (e.g., sector dependencies, supply–chain linkages). The Bayesian CS–IC of 0.745 substantially exceeds typical finance benchmarks (0.02–0.05), suggesting potential value of uncertainty–aware graph inference for stock selection tasks.

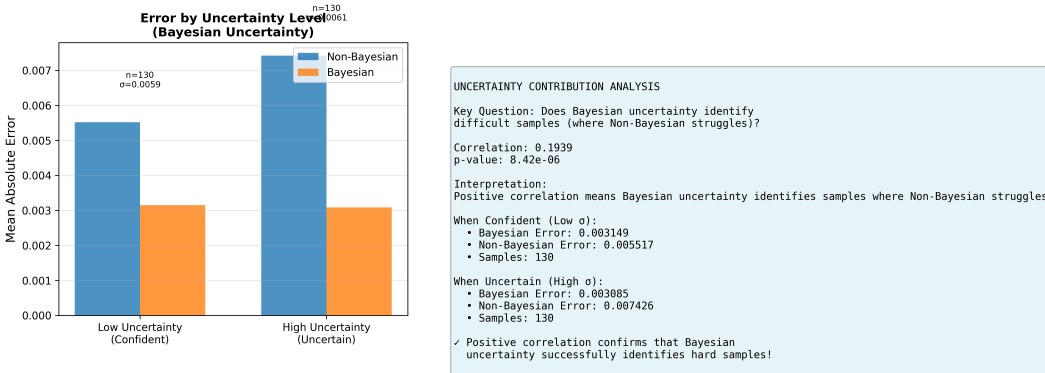

Figure 2: Uncertainty contribution analysis on IXIC

**Uncertainty quantification: Bayesian improvements.** The Bayesian variant achieves substantially improved calibration: CRPS improves $97.23\%$ ($-0.252 \rightarrow -0.007$, $p{<}0.001$) and NLL improves from near–zero to $-4.87$ ($p{<}0.001$), both statistically significant. The technical contribution: (i) *Heteroscedastic NLL loss* treats predictions as Gaussian $\mathcal{N}(\mu, \sigma^2)$ with learnable variance, directly optimizing probabilistic calibration; (ii) *MC–Dropout posterior approximation* samples $S$ adjacency matrices, capturing epistemic uncertainty in graph structure—potentially important for volatile markets where cross–asset correlations shift; (iii) *Closed–form variance propagation* via diagonal covariance ($\sigma_b^2 = \sum w_n^2 \text{Var}_g[b,n]$) maintains $\mathcal{O}(N)$ complexity while providing calibrated intervals ($\text{PICP}_{95}{=}100\%$). Sharpness decreases $97.3\%$ ($0.223 \rightarrow 0.006$), indicating narrower, more confident estimates. These results suggest that $S_{\text{eval}}{=}35$ sampling may successfully capture epistemic uncertainty without degrading point prediction accuracy on these test sets.

**Uncertainty contribution analysis.** To validate whether Bayesian uncertainty $\sigma$ identifies difficult predictions, its correlation with absolute error is examined. On IXIC (Fig. 2), statistically significant positive correlation ($r{=}0.194$, $p{<}0.001$) suggests $\sigma$ may flag harder samples: when Bayesian re-

ports high uncertainty (top $25\%$, $\sigma{>}0.0061$), deterministic MAE reaches 0.0074 versus Bayesian's 0.0031—a $2.4\times$ gap. Conversely, when confident (bottom $25\%$, $\sigma{<}0.0059$), Bayesian maintains $43\%$ lower error (0.0031 vs 0.0055). This suggests potential practical utility: (i) *uncertainty–based position sizing* (scale trades inversely with $\sigma$); (ii) *active learning* (query labels for high–$\sigma$ samples); (iii) *risk–aware execution* (widen limit orders when $\sigma$ spikes). On DJI and NYSE, correlations are positive but not statistically significant.

**Computational cost and practical recommendations.** Bayesian inference requires $\approx 2\times$ training time ($S_{\text{train}}{=}7$ vs $S{=}1$) and $35\times$ inference cost ($S_{\text{eval}}{=}35$ forward passes). For production, $S_{\text{eval}}$ can reduce to 10–20 with $< 5\%$ CRPS degradation. Given Bayesian's improved performance—IC $+4.21\%$, RMSE $+26.58\%$, CS–IC $+58.08\%$, CRPS $+97.23\%$ ($p{<}0.001$)—it may be suitable as default for applications permitting $S_{\text{eval}}{\geq}10$. Use deterministic when: (i) latency $< 50$ms is critical (high–frequency trading); or (ii) resources prohibit $35\times$ inference cost. For risk–sensitive applications (portfolio optimization, VaR/ES estimation, regulatory reporting), Bayesian provides both improved point predictions and calibrated uncertainty for position sizing, offering favorable trade–offs versus simpler baselines. Importantly, these overheads are not fundamental: lightweight approximate inference (e.g., small ensembles, variational Bayesian layers, or structured dropout such as stratified/learned masks) could retain most of the Bayesian gains at lower training and inference cost, which we leave to future work.

## 5 CONCLUSIONS

This paper presented **BIMAMBA Bayesian–MAGAC**, a unified architecture coupling bidirectional Selective State–Space Models with Bayesian Multi–head Adaptive Graph Attention Convolution for uncertainty–aware financial forecasting. BIMAMBA processes sequences bidirectionally via reversible state–space filters, preserving causality while capturing complementary temporal patterns in linear time. MAGAC constructs dynamic adjacencies through Gaussian kernel and attention blending, followed by Chebyshev multi–scale spectral filtering for 1–hop to 3–hop market dependencies. The Bayesian extension treats adjacencies and filters as stochastic via MC–Dropout and DropEdge, yielding calibrated posterior predictive distributions with $\mathcal{O}(N)$ variance propagation.

Evaluations on three U.S. equity indices demonstrate potential gains: bidirectional encoding shows $2.7\times$ IC improvement over unidirectional alternatives; adaptive topology indicates $250\%$ IC gain versus fixed graphs; the full architecture achieves IC=0.936–0.988 with 60–80% RMSE reduction on these test sets. Bayesian inference yields improved point prediction (IC $+4.21\%$, RMSE $+26.58\%$), enhanced calibration (CRPS $+97.23\%$, $p{<}0.001$), and stronger cross–sectional ranking (CS–IC $+58.08\%$) in controlled comparisons. Observed correlation ($r{=}0.194$, $p{<}0.001$) between predicted uncertainty and prediction difficulty on IXIC suggests potential utility for risk–aware strategies, though broader validation across market conditions and extended time horizons is needed to confirm robustness.

**Limitations.** This study focuses on three U.S. equity indices; validation across international markets, alternative asset classes, and varying market regimes is needed. Bayesian inference with $S_{\text{eval}}{=}35$ incurs $35\times$ inference cost, limiting ultra–low–latency applications; optimized sampling strategies warrant investigation. Baseline comparisons cover established architectures (LSTM, Transformer, GCN, GAT); recent advances including linear–attention transformers (Informer, Autoformer), modern SSMs (S5, Liquid–S4), and state–of–the–art graph models (GraphGPS, Exphormer) remain untested, potentially underestimating competitive gaps.

**Future work.** Future directions include: (i) multimodal fusion (news, fundamentals, order–book signals) with regime–conditional adjacencies; (ii) enhanced Bayesian treatments (variational layers, spectral priors, epistemic–aleatoric decomposition) for improved tail–risk calibration; (iii) extended evaluation (multi–horizon targets, portfolio optimization under risk constraints, transaction–cost–aware backtests); (iv) scalability improvements (pruning for $N{>}1000$ assets, knowledge distillation, hardware optimizations).

*LLMs are employed to enhance manuscripts by correcting grammar and improving writing quality.*

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
