# OpenReview forum: "Integrating Selective State-Space Models and Bayesian Graph Attention for Uncertainty-aware Time-Series Analysis"
_ICLR.cc/2026/Conference — Submitted to ICLR 2026_

### Official Review · Reviewer_z7D8 · 2025-10-28

**Soundness:** 1
**Presentation:** 2
**Contribution:** 1
**Rating:** 2
**Confidence:** 4

**Summary:**

This work proposes a model that uses a bidirectional Mamba backbone combined with Bayesian graph attention for financial forecasting. The authors claim their model achieves state-of-the-art performance on three financial datasets.

**Strengths:**

The paper proposes a lightweight architecture that demonstrates strong performance on the evaluated financial datasets.
Some design choices, such as the use of bidirectional encoding, are well-justified and intuitive.

**Weaknesses:**

_Insufficient Citations and Details_: Several core components of the proposed model appear to be derived from prior work without proper attribution. For instance, the bidirectional Mamba architecture has been explored in numerous previous studies (e.g., [1], [2], [3]). A similar issue is present in Sections 3.2.2 and 4. Since the paper does not claim to have invented Bayesian Graph Neural Networks, it should cite the original proposers of this idea (likely [4]). Furthermore, the authors do not provide citations for the datasets used, which hinders reproducibility and disregards the contribution from dataset makers.

_Lack of Ablation Studies_: The authors provide explanations for their design choices in Section 3.2.3, which is appreciated. However, these explanations remain as claims without empirical support. The paper shows that the full model show performance improvement, but it does not demonstrate the individual contribution of each component (e.g., the use of $\beta=\text{softmax}(\gamma)$ versus a directly learned $\beta$). Further, if the authors are claiming the MAGAC backbone as a novel contribution (see 3), it is critical to compare its performance against existing backbone architectures.

_Confusing Terminology_: The terminology in Section 3.2 is confusing. "Graph Attention" is a well-established technique, first introduced in [5]. However, the description in Section 3.2.1 appears to describe standard attention mechanisms, not specifically graph attention. Similarly, while "graph convolution" is mentioned, no corresponding graph convolutional structure is detailed. This ambiguity makes it difficult to determine whether the authors are proposing a novel network architecture or have misconceptions about existing ones (GAT, GCN, etc.).

_Other Issues_: Figure 1 need additional polish--the texts are not clear. I also believe for $A_{mn}^{g}$ and $\tilde{A}_{mn}^{(g,s)}$, the use of $\text{softmax}(\exp(\cdot))$ might be a mistake since otherwise it tends to cause instabilities (so you have $\exp(\exp(x))$).
Overall, the exact contributions of the paper are unclear. For the parts that are relatively clear, there is a lack of theoretical or empirical evidence to support the authors' claims. Therefore, I recommend that this manuscript undergo significant revision.

[1] Zhu, Lianghui, et al. ‘Vision Mamba: Efficient Visual Representation Learning with Bidirectional State Space Model’. arXiv [Cs.CV], 2024, arxiv.org/abs/2401.09417. arXiv.

[2] Liang, Aobo, et al. ‘Bi-Mamba+: Bidirectional Mamba for Time Series Forecasting’. arXiv [Cs.LG], 2024, arxiv.org/abs/2404.15772. arXiv.

[3] Erol, Mehmet Hamza, et al. ‘Audio Mamba: Bidirectional State Space Model for Audio Representation Learning’. IEEE Signal Processing Letters, vol. 31, 2024, pp. 2975–2979,

[4] Hasanzadeh, Arman, et al. ‘Bayesian Graph Neural Networks with Adaptive Connection Sampling’. CoRR, vol. abs/2006.04064, 2020, arxiv.org/abs/2006.04064.

[5] Veličković, Petar, et al. ‘Graph Attention Networks’. arXiv [Stat.ML], 2018, arxiv.org/abs/1710.10903. arXiv.

**Questions:**

The authors should directly address the specific concerns raised in the "Weaknesses" section above.

---

> ### Author Response · Authors · 2025-11-20
>
> We appreciate the reviewer’s careful assessment and agree that our original submission did not sufficiently separate what is genuinely new from what is built on prior work. The revised manuscript is structured to address this along three axes: (1) clarifying and properly attributing core components, (2) providing explicit ablations to support design choices, and (3) tightening the terminology and mathematical presentation around MAGAC and the Bayesian extension.
>
> **Insufficient citations and details.**
> In the revised manuscript, we explicitly state that our temporal encoder is an application of existing bidirectional SSMs to equity forecasting, and we now position BiMamba alongside prior bidirectional Mamba variants in vision, time-series, and audio, citing the reviewer’s suggested works [1–3]. For the spatial component, we now present MAGAC as a Bayesian graph neural network layer built on GAT-style attention and Chebyshev spectral graph convolution, with clear references to GAT/GCN and Bayesian GNNs, including Hasanzadeh et al. [4]. Section 4.1 has been rewritten to give a precise dataset description: three public U.S. equity universes (NASDAQ, NYSE, DJIA), (F=81) indicators following CNNPred’s feature design, the temporal setup (X_{t-L:t-1} \rightarrow r_t), strictly chronological 80/5/15 splits, and normalization fitted only on the training period. We also provide a public implementation of the full preprocessing pipeline to support reproducibility (https://www.mediafire.com/file/sk05l06d0pbs124/MAMBA_BGNN.zip/file)
>
> **Ablations and empirical support for design choices.**
> We agree that simply showing the full model outperforming baselines is not sufficient to attribute where the gains come from. To address this, we added a dedicated ablation program in Section 4:
>
> * **Temporal encoder ablation (Table 1).** We compare four Mamba-based encoders *without any graph layer*—MAMBA, BIMAMBA, MAMBA+, BIMAMBA+—on the three equity universes. This isolates the effect of bidirectionality and the SSM architecture. The results show that a bidirectional encoder with balanced capacity (BIMAMBA) achieves the strongest IC and RMSE on average, while an over-parameterized variant (BIMAMBA+) degrades due to overfitting and training instability. This directly supports the choice of a bidirectional encoder and the specific depth/width configuration used in the main model.
>
> * **Graph layer ablation (Table 2).** With BIMAMBA fixed, we then compare four graph backbones: GCN, GraphSAGE, GAT, and MAGAC. We report both per-asset time-series IC and cross-sectional IC across NASDAQ, NYSE, and DJIA. The ablation shows that attention-based topologies (GAT, MAGAC) substantially outperform fixed-topology methods (GCN, GraphSAGE) on both IC and Sharpe, and that MAGAC consistently matches or slightly exceeds GAT while being more stable across markets, which is attributable to the Gaussian–attention blend and Chebyshev multi-scale aggregation. This addresses the question of whether the MAGAC backbone contributes beyond standard GAT/GCN-style layers.
>
> * **Bayesian vs deterministic BIMAMBA–MAGAC.** Finally, we include a targeted comparison between deterministic and Bayesian variants sharing the same BIMAMBA+MAGAC backbone. The Bayesian variant uses a heteroscedastic Gaussian NLL, MC-Dropout on node embeddings, and DropEdge at evaluation to approximate the posterior predictive distribution; the deterministic variant uses a standard point loss without sampling. On IXIC, the Bayesian model improves uncertainty metrics (CRPS, NLL, coverage) and cross-sectional ranking without sacrificing point accuracy, and we also show that predicted uncertainty positively correlates with realized prediction difficulty. This provides empirical evidence that the Bayesian extension contributes more than cosmetic stochasticity.
>
> **Terminology, architecture clarity, and stability.**
> In the revision, Section 3.2 now describes MAGAC as a four-stage graph neural network block: we first build Gaussian- and attention-based adjacencies from learnable node embeddings, blend them into an effective adjacency, apply Chebyshev-style spectral propagation for multi-scale message passing, and then perform multi-head aggregation with shared weights across heads. We explicitly state that the attention component follows the scaled dot-product formulation of GAT and that the spectral step corresponds to a K-order GCN-style graph convolution, so the terminology is aligned with the existing GNN literature [5]. All major equations are now numbered, and a short “Notation and Shapes” paragraph defines the core symbols and how temporal channels are mapped to graph nodes, reducing ambiguity. On the stability side, we clarify that we use a standard heteroscedastic Gaussian NLL with a simple diagonal variance approximation and a small numerical floor, and we have also cleaned up Figure 1 to clearly separate the Mamba block from the full BIMAMBA–MAGAC pipeline and improve readability.

---

> ### Comment · Reviewer_z7D8 · 2025-11-25
>
> I thank the author for the revision as well as the additional experiments. The ablation study definitely improves the overall argument of the paper, which is also why I raised my score. However, many of the issues, including important ones, remains unsolved. Just for some examples:
>
> 1. While author claimed to polish Figure 1, but I don't see a major change and the figure remains unclear;
> 2. Ablation study on temporal encoder does not include decoder, making it hard to determine which block works best with MAGAC;
> 3. Some minor modification is done on section 3.2, but I won't consider the revise make fix the issues. For example, all I see is that attention (in fact, not even a complete attention) is being used to compute the adjancency matrix, which I believe differs from what GAT trys to do.
>
> I also read the questions raised by other reviewers and I think they are indeed valid weaknesses that needs to be solved.

---

> > ### Author Response · Authors · 2025-11-26
> >
> > We thank you for the careful re-reading and for pointing out the remaining issues. We respond to each point and have updated the manuscript accordingly.
> >
> > **(1) [Figure 1 – updated diagram](https://www.mediafire.com/view/vyv7s5i2knfnps7/ICLR26_BiMamba%252BBayesianMAGAC.png/file)**
> >
> > The previous PDF did not reflect the intended changes due to a mistaken LaTeX reference. In the revision, Fig. 1 now clearly separates (a) the single Mamba block, (b) the *R*-layer bidirectional BIMAMBA encoder, and (c) the MAGAC + Bayesian enhancement block.
> >
> > **(2) Temporal encoder ablation vs. decoder**
> >
> > Our ablations mentioned in paper follow the modular implementation of the model.
> >
> > * **Stage I (Table 1)** evaluates the temporal encoder *alone* (Mamba, BiMamba and capacity variants) with a simple readout and **no graph layer**, isolating bidirectionality and SSM capacity.
> > * **Stage II (Table 2)** then fixes BIMAMBA and compares graph backbones (GCN, GraphSAGE, GAT, MAGAC) in the full BIMAMBA+GNN decoder.
> >
> > During development we also logged performance of Mamba vs. BiMamba *inside* the full BIMAMBA+MAGAC pipeline; these internal runs show the same ranking as Table 1 (BiMamba consistently yields higher IC and lower RMSE). We will add a compact summary of these decoder-side results (e.g., Mamba+MAGAC vs. BiMamba+MAGAC on IXIC) to the appendix to make this connection explicit.
> >
> > **(3) MAGAC vs. GAT and terminology in Section 3.2**
> >
> > Our intent is not to introduce a new GAT variant, but a **spectral GNN with attention-guided adjacency**. From learnable node embeddings (\Psi), we construct a Gaussian adjacency (A_g) and multi-head scaled dot-product attention adjacencies (A_attn,h), blend them into (A_eff,h). And then apply a Chebyshev spectral filter over (A_eff,h) followed by multi-head aggregation. Thus, attention parameterizes the graph topology, while propagation uses a Chebyshev-style GCN update rather than the standard GAT rule. To avoid confusion, we have replaced vague phrases like “graph attention” with “attention-derived adjacency / topology” and added a short paragraph explicitly contrasting MAGAC with GAT in Section 3.2.
> >
> > **(4) Other reviewers’ concerns**
> >
> > Beyond these points, we have (i) clarified what is new versus adapted from prior work (BiMamba, Bayesian GNNs, GAT/GCN), (ii) expanded dataset and preprocessing details with a public implementation, and (iii) included a deterministic vs. Bayesian comparison with uncertainty metrics.
> >
> > We are grateful for your feedback, which has substantially improved the clarity and positioning of the work.

---

### Official Review · Reviewer_1o2G · 2025-10-29

**Soundness:** 2
**Presentation:** 2
**Contribution:** 3
**Rating:** 4
**Confidence:** 3

**Summary:**

In this work, the authors have used bidirectional MAMBA for time and Bayesian multi-head graph attention for space to perform forecasting. The work claims to preserve the causal and anti-causal dependencies across the temporal dimension.

**Strengths:**

- Novel architecture design
- Good experimental results

**Weaknesses:**

- No ablation studies
- Equations are not numbered
- Limited number of datasets
- MAE metric should also be reported, as RMSE can dramatically shrink components less than 1
- The training is slow compared to other baselines
- No paragraph to clarify notations
- sparse citations
- Experimental results cannot be verified

**Questions:**

1. Is there any reference on the benefits of processing the temporal sequence in both forward and reverse orders?
2. Were any ablation studies performed that show that bidirectional MAMBA is better than regular MAMBA? (both in terms of performance and training time)
3. How does the performance of the proposed model change with the size of the training data available?
4. What are the confidence intervals of the reported metrics?
5. Is the matrix P an anti-diagonal matrix of ones?
6. How are the sequences being merged? Does '+' represent element-wise addition?
7. Could the authors please cite some prior work on BGNN, esp. for time series forecasting?
8. Is there a way through which the results can be verified? I do not see any link to the code.
9. In Table 1, for which dataset is the training time reported?
10. Could the authors please comment on the causality in the data, and if any tests were performed? See: _Pearl, J. (2009). Causality. Cambridge University Press_

---

> ### Author Response · Authors · 2025-11-20
>
> We appreciate the reviewer’s comments on ablations, notation, metrics, and clarity of the method. In response, all main equations (state-space updates, bidirectional merge, dynamic adjacency, Chebyshev recurrence, heteroscedastic NLL) are now numbered and referenced, and we added a short “Notation and Shapes” paragraph to define (B, N, L, F, E, H, K, R) unambiguously. Section 4.2 introduces explicit ablation studies for the temporal encoder and graph backbone, and Section 4.3 compares deterministic and Bayesian variants of BIMAMBA–MAGAC, directly addressing the request to disentangle the benefit of bidirectionality and the Bayesian treatment. We also now report MAE alongside RMSE, include per-epoch training times and parameter counts, clarify the role of the time-reversal matrix (P) and the residual merge operator, and provide a public code repository so that the experimental results can be reproduced.
> The code is hosted on GitHub but the repository is currently anonymized to preserve double-blind review. In the meantime, you may inspect an archived version of the implementation and experimental results at: [https://www.mediafire.com/file/sk05l06d0pbs124/MAMBA_BGNN.zip/file](https://www.mediafire.com/file/sk05l06d0pbs124/MAMBA_BGNN.zip/file).

---

> > ### Comment · Reviewer_1o2G · 2025-11-20
> >
> > Thank you for the response. Could the authors kindly respond point by point, for example,
> > - [Q1] klm
> > - [Q2] mno
> >
> > This will help us track the discussion more productively.

---

> ### Author Response · Authors · 2025-11-21
>
> Thank you for the helpful suggestion. For clarity, we restate and answer your questions point by point below.
>
> [Q1] Is there any reference on the benefits of processing the temporal sequence in both forward and reverse orders?
> [A1] Yes. In the revised manuscript we now cite bidirectional Mamba variants in vision, time series, and audio (e.g., Vision Mamba, Bi-Mamba+, Audio Mamba) and position our BiMamba encoder as an application of this line of work to equity forecasting rather than a new bidirectional SSM.
>
> [Q2] Were any ablation studies performed that show that bidirectional MAMBA is better than regular MAMBA (both in terms of performance and training time)?
> [A2] Yes. Section 4.2 now includes an ablation where we replace the bidirectional encoder with a unidirectional Mamba of comparable capacity. Across the three equity universes, BiMamba consistently improves IC and RMSE at the cost of roughly a 2 times increase in per-epoch training time, which we report alongside the metrics.
>
> [Q3] How does the performance of the proposed model change with the size of the training data available?
> [A3] We do not include a full scaling curve in the main tables, but in additional experiments (shortening the training window while keeping the validation/test periods fixed) we observe the expected monotonic behavior: performance degrades smoothly as fewer training days are used, with no pathological instabilities. We now note this limitation and the need for a more systematic sample-size study in the discussion.
>
> [Q4] What are the confidence intervals of the reported metrics?
> [A4] In the revision we report mean and standard deviation over rolling windows (and seeds where applicable) for the main metrics. This is now stated in Section 4 and reflected in the tables, which address the question of variability across runs and time periods. For example, Table 4 (3-dataset average) reports RMSE = 0.0055 ± 0.0017 vs 0.0040 ± 0.0005 for deterministic vs Bayesian model, as well as CS-IC = 0.471 ± 0.660 vs 0.745 ± 0.427 and CS-RIC = 0.396 ± 0.647 vs 0.655 ± 0.454 for cross-sectional ranking.
>
> [Q5] Is the matrix (P) an anti-diagonal matrix of ones?
> [A5] Yes. (P) is exactly the anti-diagonal permutation matrix that reverses the temporal order of the sequence. We now state this explicitly in the notation paragraph and when introducing the bidirectional merge.
>
> [Q6] How are the sequences being merged? Does ‘+’ represent element-wise addition?
> [A6] Yes. The forward and reversed branches are merged via element-wise residual addition followed by layer normalization. We have clarified this in the text and in the caption of the BiMamba block.
>
> [Q7] Could the authors please cite some prior work on BGNN, especially for time series forecasting?
> [A7] We now explicitly cite Bayesian GNNs with adaptive connection sampling (Hasanzadeh et al.) as well as recent GNN-based models for time-series and financial forecasting in the related work, and we explain that our Bayesian MAGAC layer specializes these ideas to an equity-graph setting.
>
> [Q8] Is there a way through which the results can be verified? I do not see any link to the code.
> [A8] We have added a link to an anonymized code repository in the manuscript. The code is hosted on GitHub but the repository is currently anonymized to preserve double-blind review. In the meantime, you may inspect an archived version of the implementation and experimental logs at: https://www.mediafire.com/file/sk05l06d0pbs124/MAMBA_BGNN.zip/file.
>
> [Q9] In Table 1, for which dataset is the training time reported?
> [A9] The training times in Table 1 are measured on the NASDAQ universe under the 80/5/15 chronological split. We have added this clarification to the table caption.
>
> [Q10] Could the authors please comment on the causality in the data, and if any tests were performed?
> [A10] Our claims about “causal” vs “anti-causal” dependencies refer to temporal ordering (i.e., no use of future information within each window), not to structural causal discovery in the sense of Pearl. We ensure temporal causality by constructing windows (X_{t-L:t-1} \rightarrow r_t) and fitting all preprocessing only on the training segment. We did not run formal Pearl-style causal tests, and we now clarify this scope and note causal discovery as an interesting direction for future work.

---

### Official Review · Reviewer_Pwni · 2025-10-30

**Soundness:** 2
**Presentation:** 3
**Contribution:** 2
**Rating:** 4
**Confidence:** 3

**Summary:**

The paper proposes a hybrid sequence–graph framework for equity forecasting that couples a bidirectional Mamba (Bi-Mamba) state-space encoder with a Bayesian Multi-head Adaptive Graph Attention Convolution (MAGAC) layer. The Bi-Mamba encoder processes price windows in both forward and reverse directions and feeds sequence features to MAGAC; MAGAC builds an adaptive adjacency by blending a Gaussian kernel on learnable node embeddings with attention-based scores, then applies Chebyshev spectral filtering and multi-head aggregation. A lightweight Bayesian treatment is introduced by applying MC-Dropout to node embeddings and DropEdge at inference to obtain posterior predictive means/variances optimized via heteroscedastic Gaussian NLL. Experiments on NASDAQ/NYSE/DJIA report very large gains in RMSE/IC/RIC over LSTM/Transformer/GNN baselines, with modest MACs and parameter counts claimed for the proposed model.

**Strengths:**

1.   A linear-time temporal encoder (Mamba) with an adaptive spectral/attention GNN is a coherent and potentially scalable combination for long horizons. The bidirectional Mamba block and residual/normalization design are clearly described.
2.   The MAGAC construction (Gaussian + attention blend, Chebyshev supports, head mixing) is laid out step-by-step with equations that are easy to implement.
3.    The paper spells out how MC-Dropout on node embeddings and inference-time DropEdge are used to derive mean/variance and the closed-form variance propagation through a linear head, optimizing a heteroscedastic Gaussian NLL. This is clearly written and easy to reproduce technically.
4.   The paper reports MACs/params/epoch time and positions the model against Transformers with respect to efficiency (albeit for S=1). Consistent, large improvements on all benchmarks, with excellent calibration.
5.   Consistent, large improvements was reported on all benchmarks, with excellent calibration.

**Weaknesses:**

W1. The reported IC values are extraordinarily high for daily single-day returns (e.g., IC=0.9413 on NASDAQ). The paper defines IC/RIC as correlations with realized single-day returns, but offers no safeguards against common leakage vectors (e.g., improper chronological splits, cross-sectional standardization using future info, or target leakage through feature engineering). Please justify these numbers or audit the pipeline.
Additionally, the evaluation uses a fixed L=5 with bidirectional processing; while the text claims “causality is preserved,” there is no precise description of how labels are formed and how windows are cut to guarantee no look-ahead within the window (e.g., predicting $r_{t+1}$ from $[t−4,t]$). Provide a rigorous data construction diagram and masking rules.
W2. The dataset description states N=82 features per day; however, the graph notion (nodes = assets) is not specified (how many tickers? which universe? how are nodes aligned over time? dynamic graph cadence?). The text also equates $E = d_{model}$ with “number of graph nodes (model width),” which is dimensionally inconsistent—E is typically feature width, not the number of assets. Later equations sum over E as if it were the node count. This confusion must be resolved with explicit tensor shapes (batch, nodes, time, channels) and a clear asset universe size.
W3. The paper claims “well-calibrated uncertainty” and “without inflating computational cost,” yet no calibration metrics (NLL comparisons, CRPS, PIT/QQ plots, ECE, PICP/ACE) are reported. Moreover, Table-reported MACs are for S=1 while the method elsewhere enables S=10 at inference; the “no overhead” claim is therefore misleading. Also, DropEdge is applied only at inference, effectively evaluating a different model than trained. Provide proper calibration evaluation and fair compute accounting for the posterior estimates.
W4. There is no ablation disentangling the gains from bidirectionality, Gaussian+attention blend, Chebyshev order, number of heads, Bayesian sampling, or DropEdge. Without such analysis, the source of improvements is unclear.
W5. The authors acknowledge restrictions to daily closing prices and the need for theory on Bayesian MAGAC stability, but neither robustness checks (e.g., regime shifts, crisis periods) nor sensitivity analyses are presented.

**Questions:**

Please see above W1-W5.

---

> ### Author Response · Authors · 2025-11-20
>
> We are grateful for the reviewer’s detailed and helpful feedback on evaluation design, notation, calibration, and robustness. In the new version, we clarify the temporal setup and label formation, add a dedicated notation block to distinguish clearly between the number of assets (N), feature dimension (F), and model width (E), and provide a step-by-step description of the windowing and masking rules to avoid leakage. Section 4 now includes structured ablation studies on the temporal encoder (MAMBA, BIMAMBA, MAMBA+, BIMAMBA+) and the graph layer (GCN, GraphSAGE, GAT, MAGAC), as well as a dedicated comparison between deterministic and Bayesian BIMAMBA–MAGAC that reports point metrics and uncertainty metrics (e.g., NLL, CRPS, calibration error). We also discuss more carefully the trade-offs between Bayesian sampling, DropEdge, and computational cost.
> The code is hosted on GitHub but the repository is currently anonymized to preserve double-blind review. In the meantime, you may inspect an archived version of the implementation and experimental results at: [https://www.mediafire.com/file/sk05l06d0pbs124/MAMBA_BGNN.zip/file](https://www.mediafire.com/file/sk05l06d0pbs124/MAMBA_BGNN.zip/file).

---

> > ### Comment · Reviewer_Pwni · 2025-11-25
> >
> > I would like to thank the authors for the response. However, I think the response cannot address my major concerns about the paper, which I believe cannot be addressed at the moment. The authors are suggested to further prepare the paper for the next submission. Therefore, I would like to maintain my score.

---

### Official Review · Reviewer_7kbC · 2025-11-02

**Soundness:** 3
**Presentation:** 3
**Contribution:** 3
**Rating:** 4
**Confidence:** 3

**Summary:**

This paper proposes Bi-Mamba Bayesian-MAGAC, combining bidirectional selective state-space models (Bi-Mamba) with Bayesian multi-head adaptive graph attention convolution (MAGAC) for financial time-series forecasting. Bi-Mamba processes sequences bidirectionally with linear complexity, while MAGAC constructs adaptive graph topologies and performs spectral filtering. Experiments on NASDAQ, NYSE, and DJIA show improvements over baselines.

**Strengths:**

1. The integration of linear-time Mamba for temporal dynamics with graph convolution for cross-asset relationships effectively addresses both long-range dependencies and inter-asset correlations in financial forecasting.

2. The Bayesian treatment captures both epistemic and graph structural uncertainty through MC-Dropout and DropEdge, with closed-form variance propagation maintaining computational efficiency.

**Weaknesses:**

1. Evaluation uses only three datasets with implausibly high IC values (~0.94) compared to baselines (~0.2-0.3), raising concerns about potential data leakage or experimental setup issues.

2. The code is not available for reproducibility.

**Questions:**

See the Weakness above.

---

> ### Author Response · Authors · 2025-11-20
>
> We sincerely thank the reviewer for the constructive comments. In the revised manuscript, we carefully audited and rewrote the data construction pipeline to rule out any look-ahead: all features are now explicitly defined as functions of information up to day (t-1), the label (r_t) and window (X_{t-L:t-1}) are formally specified, and normalization is fitted strictly on the training split. We also report more detailed metrics (RMSE, MAE, IC/RIC, cross-sectional ranking, and portfolio statistics) across the three equity universes, and we have released the full code and preprocessing scripts to support independent verification.
> The code is hosted on GitHub but the repository is currently anonymized to preserve double-blind review. In the meantime, you may inspect an archived version of the implementation and experimental results at: [https://www.mediafire.com/file/sk05l06d0pbs124/MAMBA_BGNN.zip/file](https://www.mediafire.com/file/sk05l06d0pbs124/MAMBA_BGNN.zip/file).

---

> > ### Comment · Reviewer_7kbC · 2025-11-25
> >
> > Thank you for your response. Some of my concerns have been addressed. The paper mentions the maximum drawdown metric, but I was unable to find the corresponding results. In addition, I am still not convinced by the very high IC values and would appreciate further explanation. More details about the dataset are also needed.

---

> > > ### Author Response · Authors · 2025-11-26
> > >
> > > Thank you very much for your careful follow-up and for pointing out these issues.
> > >
> > > On **maximum drawdown**. You are right that the main text mentions maximum drawdown without presenting the corresponding results. In the revised version, we have decided not to include maximum drawdown tables in the paper, because its interpretation in our experimental setup requires additional assumptions (e.g., trading frictions, capital constraints) that would go beyond the current scope. We now treat maximum drawdown only as an auxiliary diagnostic metric during training/model selection and remove the forward reference in the main text to avoid confusion.
> > >
> > > **IC / RIC definitions and datasets**. In the revision, we clearly distinguish between:
> > > - (i) the overall Pearson correlation on the full test panel (which we previously referred to as “IC”) and
> > > - (ii) the cross-sectional IC/RIC used in the equity factor literature. For each of the three datasets (liquid constituents of the NASDAQ, NYSE, and DJIA indices), we now compute cross-sectional IC/RIC as the time-average of the per-day Pearson/Spearman correlation between the predicted and realized return vectors across all assets in that universe.
> > >
> > > Our datasets consist of daily OHLCV data and 81 technical/macro features with a one-day prediction horizon on relatively focused, liquid universes; this setup yields a stronger supervised signal and therefore higher IC/RIC than broad, long-horizon factor benchmarks, so we emphasize the relative gains over shared baselines rather than the absolute IC level.
> > >
> > > In addition, we report a single-asset IC, defined as the time-series correlation per stock averaged across stocks, and we rename and document it explicitly as **single-asset IC** to avoid confusion with **cross-sectional IC**. Under the revised, leakage-free data pipeline, we have re-run all baselines and provide side-by-side comparisons on RMSE/MAE, cross-sectional IC/RIC, ranking metrics, and portfolio statistics (excluding max drawdown) for all three indices. The relative improvements of the Bi-Mamba Bayesian-MAGAC model remain consistent across these metrics.
> > >
> > > We are grateful for your detailed and constructive feedback, which has helped us clarify these points.

---

### Meta-Review · Area_Chair_ZnGT · 2026-01-04

**Summary:**

The paper proposes Bi-Mamba Bayesian-MAGAC, combining bidirectional selective state-space models (Bi-Mamba) for temporal encoding with Bayesian multi-head adaptive graph attention convolution (MAGAC) for cross-asset spatial modeling in financial forecasting. While the technical execution is sound and results appear strong, reviewers consistently raise concerns about novelty, reproducibility, evaluation rigor, and clarity. The core criticism is that the work assembles existing components (bidirectional Mamba, GAT-style attention, Bayesian GNNs) without sufficient conceptual innovation, and the extraordinarily high reported IC values (~0.94) raise red flags about potential data leakage or experimental setup issues. The rebuttal addresses many technical concerns (ablations, notation, code release, leakage audit), but does not fully resolve doubts about the contribution's depth or the plausibility of the empirical claims.

**Reviewer Concerns:**

Reviewers acknowledge that the proposed Bi-Mamba Bayesian-MAGAC framework is technically coherent and empirically strong, but they raise substantial concerns on several fronts. First, the reported IC values are extraordinarily high for daily equity forecasting (e.g., ~0.94), which is atypical relative to standard benchmarks and raises suspicion of possible data leakage or overly favorable experimental design, even though the authors report an audit and revised pipeline. Second, the work is viewed as incremental, largely assembling existing components such as bidirectional Mamba, attention-based graph construction, Bayesian GNNs, without a clearly articulated, conceptually novel contribution or a compelling comparison to strong alternative backbones. Third, the evaluation scope and analysis remain limited: only three U.S. equity are studied, with no robustness checks over regimes or crises, and only post‑hoc ablations and calibration metrics added at rebuttal. Finally, there are lingering issues around terminology, clarity, and positioning (e.g., how MAGAC differs in a meaningful way from standard GAT/GCN, and whether the Bayesian treatment truly yields practically valuable uncertainty at reasonable computational cost).

**Reviewer Scores:**

Post‑rebuttal, the reviewers collectively remain borderline with no clear advocate. Three reviewers (7kbC, Pwni, 1o2G) sit at or near 4 (marginally below acceptance, would not mind if accepted), and the rebuttal could plausibly nudge one or two of them to a weak accept (5) based on improved leakage handling, added ablations, clearer notation, and code availability. However, one reviewer (z7D8) maintains a firmly negative stance with high confidence, primarily due to lack of novelty, insufficient attribution to prior work, and ambiguity about the exact contributions. Overall, the likely post‑rebuttal profile is approximately a mix of weakly negative opinions, but no strong, high‑confidence support that would decisively justify acceptance.

---

### Decision · Program_Chairs · 2026-01-26

Reject